# Transcriptome Analysis Reveals an Essential Role of Exogenous Brassinolide on the Alkaloid Biosynthesis Pathway in Pinellia Ternata

**DOI:** 10.3390/ijms231810898

**Published:** 2022-09-17

**Authors:** Chenchen Guo, Ying Chen, Dengyun Wu, Yu Du, Mengyue Wang, Cunqi Liu, Jianzhou Chu, Xiaoqin Yao

**Affiliations:** 1School of Life Sciences, Hebei University, Baoding 071002, China; 2Institute of Life Sciences and Green Development, Hebei University, Baoding 071002, China; 3Key Laboratory of Microbial Diversity Research and Application of Hebei Province, Baoding 071002, China; 4Hebei Key Laboratory of Wetland Ecology and Conservation, Hengshui 053000, China

**Keywords:** alkaloid, brassinolide, ephedrine, secondary metabolism, RNA-seq

## Abstract

*Pinellia ternata* (Thunb.) Druce is a traditional medicinal plant containing a variety of alkaloids, which are important active ingredients. Brassinolide (BR) is a plant hormone that regulates plant response to environmental stress and promotes the accumulation of secondary metabolites in plants. However, the regulatory mechanism of BR-induced alkaloid accumulation in *P. ternata* is not clear. In this study, we investigated the effects of BR and BR biosynthesis inhibitor (propiconazole, Pcz) treatments on alkaloid biosynthesis in the bulbil of *P. ternata*. The results showed that total alkaloid content and bulbil yield was enhanced by 90.87% and 29.67% under BR treatment, respectively, compared to the control. We identified 818 (476 up-regulated and 342 down-regulated) and 697 (389 up-regulated and 308 down-regulated) DEGs in the BR-treated and Pcz-treated groups, respectively. Through this annotated data and the Kyoto encyclopedia of genes and genomes (KEGG), the expression patterns of unigenes involved in the ephedrine alkaloid, tropane, piperidine, pyridine alkaloid, indole alkaloid, and isoquinoline alkaloid biosynthesis were observed under BR and Pcz treatments. We identified 11, 8, 2, and 13 unigenes in the ephedrine alkaloid, tropane, piperidine, and pyridine alkaloid, indole alkaloid, and isoquinoline alkaloid biosynthesis, respectively. The expression levels of these unigenes were increased by BR treatment and were decreased by Pcz treatment, compared to the control. The results provided molecular insight into the study of the molecular mechanism of BR-promoted alkaloid biosynthesis.

## 1. Introduction

*Pinellia ternata* (Thunb.) Druce is a perennial herb, and the tuber and bulbil have been widely used in traditional medicine. Pharmacological studies have revealed that *P. ternata* contains various medicinally active ingredients, such as alkaloids, lectins, volatile oils, sterols, etc. [1]. The tuber and bulbil of *P. ternata* contain a large number of alkaloids, which are the main active ingredients used for medicinal applications [2]. Driven by its wide medicinal functions, *P. ternata* has long been used to treat diseases such as cough, vomiting, inflammation, epilepsy, cancer, and traumatic injuries. However, with the increasing demand for *P. ternata*, it is becoming more difficult to obtain sufficient amounts to meet the huge demand, and over-exploitation of *P. ternata* makes these wild resources on the verge of extinction. The wild resources of *P. ternata* are widely distributed in East Asia and it is cultivated artificially in many areas in China. However, *P. ternata* rapidly withered when subjected to stresses of high light and heat under artificial cultivation, known as “sprouting tumbling”, which reduced *P. ternata* yield and quality [3]. How to improve the content of active components (especially alkaloids) of *P. ternata* is an urgent problem to be solved in artificial planting of *P. ternata*.

The secondary metabolism of plants often produces a class of nitrogenous compounds known as alkaloids. A total of 40 alkaloids, including nucleoside alkaloids, cyclic dipeptide alkaloids, and indole alkaloids, were isolated from *Pinellia species* [1]. Many alkaloids have direct pharmacological activities in medicine and modern pharmacology. The alkaloids from five processed *P. ternata* products have significant cytotoxicity against chronic myeloid leukemia cells (K562) [1]. Compared with the negative control group, the doses of 400, 200, and 100 μg mL^−1^ of *P. ternata* alkaloids inhibited the cell proliferation of human hepatocarcinoma cell strain Bel-740 by 36.98%, 15.20%, and 12.97%, respectively [4]. Oshio et al. isolated ephedrine from *P. ternata* and found that ephedrine, the same compound isolated from *P. ternata* and Ephedra species, has different uses in Chinese medicine [5]. Ephedrine alkaloids belong to the phenylpropyl-amino alkaloids and are naturally found in different plant species, such as *Catha edulis*, Ephedra species, *P. ternata*, *Roemeria refracta*, *Taxus baccata*, *Sida cordifolia*, and *Aconitum napellus* [6]. Previous studies on the ephedrine biosynthetic pathway mainly focused on the *Catha edulis* and Ephedra species plants [7,8,9]. There have been fewer studies on the ephedrine biosynthetic pathway in *P. ternata*. The study by Zhang et al. speculated a pathway for the synthesis of ephedrine from phenylalanine in *P. ternata* and identified some candidate genes involved in this process [2]. Phenylalanine is often used as precursor for the production of phenylpropyl-amino alkaloids. The first step is the formation of trans-Cinnamic acid from phenylalanine catalyzed by phenylalanine ammonia-lyase (PAL). The two possible courses of trans-Cinnamic acid to benzoic acid are β-oxidative (CoA-dependent) and non-β-oxidative (CoA-independent) routes [10]. The synthesis of secondary metabolites in plants, such as alkaloids and flavonoids, is regulated by a variety of factors. Alkaloids are usually classified as phytoalexins that help plants to resist stress conditions, and their accumulation in plants is induced by various factors, including salicylic acid, acetylsalicylic acid, and methyl jasmonate [11]. Blue light treatment improved the accumulation of alkaloids and alkaloid biosynthetic pathway genes in *Lycoris longituba* [12]. Duan et al. reported that total alkaloids and ephedrine content of in vitro cultured *P. ternata* were increased by 2.5- and 3.1-fold after 15 days of induction with 100 µL salicylic acid, compared to the control [11]. Both methyl jasmonate and ethylene had a significant effect on terpene indole alkaloid biosynthesis in the shoots and roots of *Catharanthus roseus* [13]. The galanthamine and lycorine alkaloid contents of in vitro cultures of *Leucojum aestivum* containing melatonin were 58.6- and 1.5-fold higher than those of the control group, respectively [14].

Brassinolide ((22R,23R,24S)-2α,3α,22,23-tetrahydroxy-24-methyl-B-homo-7-oxa-5α-cholestan-6-one, BR) is a plant hormone that plays a key role in plant growth and development [15]. BR improves plant tolerance to environmental stresses and thus promotes plant growth. Basit et al. reported that BR (0.01 μM) treatment alleviated the damage of chromium on rice seedlings [16]. Treatment with 24-epibrassinolide increased the photosynthetic activity, stomatal aperture, cell viability, and the activities of antioxidant enzymes and proline in tomato plants, thus enhancing the tolerance to copper stress [17]. Interestingly, BR treatment also promoted the accumulation of plant secondary metabolites. The phenolic compounds and flavonoid content in *Brassica nigra* were enhanced by 24-epibrassinolide treatment [18]. Treatment with 28-homobrassinolide enhanced the forskolin content by 73.47% in roots of *Coleus forskobliIi* [19]. In our previous study, 0.10 mg L^−1^ BR treatment increased total alkaloid content in bulbil of *P. ternata* grown in the field by 21%, compared to the control [20]. However, the mechanism of BR promoting alkaloid accumulation in bulbil of *P. ternata* is not clear and the understanding of the alkaloid synthesis pathways in *P. ternata* is poorly known. In addition, propiconazole (1-[[2-(2,4dichlorophenyl)-4-propyl-1,3-dioxolan-2-yl]methyl]-1,2,4-triazole, Pcz) is an economical and specific inhibitor of BR biosynthesis in *Arabidopsis*, maize, and soybean [21,22,23]. Therefore, understanding the regulation of BR on alkaloid biosynthesis pathways is vital to improving the quality of *P. ternata*.

The purpose of this study was to investigate the effects of BR and BR biosynthesis inhibitor (Pcz) application on alkaloid biosynthesis pathways in the bulbil of *P. ternata*. RNA-Seq was used to explore the molecular mechanism of BR promoting alkaloid accumulation. Furthermore, functional annotation and differentially expressed genes (DEGs) analysis were performed. We also identified DEGs related to ephedrine alkaloid biosynthesis, tropane, piperidine, and pyridine alkaloid biosynthesis, indole alkaloid biosynthesis, and isoquinoline alkaloid biosynthesis, and quantitative real-time PCR (qRT-PCR) was performed to verify their expression levels. Based on the transcriptome sequences and physiological data, the promoting roles of BR on alkaloid were revealed, which provided a reference for understanding the regulation of BR in alkaloid synthesis in *P. ternata*. We hypothesize that BR treatment increases the alkaloid content of bulbil of *P. ternata* by enhancing alkaloid biosynthesis.

## 2. Results

### 2.1. RNA-Seq Sequencing Analysis

To investigate the transcriptome response to BR induction in the bulbil of *P. ternata*, the nine cDNA libraries constructed with high-quality RNA from three repetitions of control, BR-treated, and Pcz-treated groups were sequenced using the Illumina Nova seq 6000 platform. These libraries generated a total of 73.14 Gb clean data, producing 57,117,606, 52,536,920, 53,506,752, 53,506,752, 50,190,552, 62,617,020, 55,852,426, 55,006,196, and 54,818,536 clear reads, respectively. The Q30 (sequencing error rate < 0.1%) was at least 94.25%. Subsequently, we obtained 115,445 unigenes with an N50 value of 1189 bp by assembling clean reads. The length range of these unigenes was 201 to 18826 bp, with an average length of 754 bp (Figure 1A). As shown in Appendix A, functional annotation results showed that 40,992 unigenes (35.51% of total unigenes) were annotated in the six databases. In the GO database, the functional classification of the 33,092 unigenes were grouped into cellular process (14,500 genes, 31.93%), metabolic process (13,235 genes, 29.14%), biological regulation (4933 genes, 10.86%), response to stimulus (3869 genes, 8.52%), localization (2885 genes, 6.35%), cellular component organization biogenesis (2585 genes, 6.35%), etc. (Figure 1B). In the KEGG database, there are 15,065 unigenes annotated to the classification of biological structure function, such as metabolism, cellular process, organismal systems, etc., and the uppermost categorization was metabolism (4209 genes) and genetic information processing (2690 genes) (Figure 1C).

### 2.2. Comparative Analysis of DEGs

A total of 818 (476 up- and 342 down-regulated) and 697 (389 up- and 308 down-regulated) DEGs were identified in BR-treated and Pcz-treated groups, respectively (Figure 2A,B). In order to further investigate the DEGs in the BR-treated and Pcz-treated groups, we performed GO and KEGG pathway enrichment analyses. The results of GO enrichment analysis showed that plant-type cell wall (GO: 0009505), secondary cell wall (GO: 0009531), cell wall (GO: 0005618), polysaccharide catabolic process (GO: 0000272), catabolic process (GO: 0009056), negative regulation of growth (GO: 0045926), etc. were significantly enriched under BR treatment (*p* < 0.05) (Figure 2C). Fourteen enriched GO terms, terpene synthase activity (GO: 0010333), hemicellulose metabolic process (GO: 0010410), cellular carbohydrate metabolic process (GO: 0044262), cell wall polysaccharide metabolic process (GO: 0010383), etc. were found in Pcz treatments (Figure 2D). As shown in Figure 2E, the phenylpropanoid biosynthesis (map 00940), pentose and glucuronate interconversions (map 00040), starch and sucrose metabolism (map 00500), MAPK signaling pathway-plant (map 04016), plant hormone signal transduction (map 04075), brassinosteroid biosynthesis (map 00905), etc. were enriched in the control vs. BR group, and the sesquiterpenoid and triterpenoid biosynthesis (map 00909), brassinosteroid biosynthesis (map 00905), phenylpropanoid biosynthesis (map 00940), MAPK signaling pathway-plant (map 04016), plant hormone signal transduction (map 04075), phenylalanine, tyrosine and tryptophan biosynthesis (map 00400), etc. pathways were enriched in the control vs. Pcz group (Figure 2F). In particular, the phenylpropanoid biosynthesis (map 00940) was significantly enriched in BR-treated and Pcz-treated groups (*p* < 0.05).

### 2.3. BR Treatment Improves Ephedrine Biosynthesis

BR treatment improved the total alkaloid content and bulbil yield by 90.87% and 29.67%, respectively, compared to the control (Figure 3A,B). There was no significant effect of Pcz treatment on total alkaloid content in bulbil. Conversely, the bulbil yield was decreased by 16.17% under Pcz treatment, compared to the control. As reported in the study by Zhang et al., the possible pathways of ephedrine biosynthesis in *P. ternata* were shown in Figure 3C [2]. Heatmap analysis shows the expression levels of twenty-six unigenes involved in the ephedrine biosynthesis in *P. ternata* bulbil (Figure 3D). These unigenes were annotated as PAL (3 genes), 4-coumaroyl-CoA ligase (4CL, 10 genes), 3-hydroxyisobutyryl-CoA hydrolase (CHY, 3 genes), 3-ketoacyl-CoA-thiolase (KAT, 5 genes), acetolactate synthase (AHAS, 1 gene), and ThDP-dependent pyruvate decarboxylase (ThDPC, 4 genes). The expression levels of two PAL (DN24895_c0_g1 and DN33061_c1_g1), three CHY (DN7851_c0_g1, DN55716_c0_g1, and DN3831_c1_g1), five KAT (DN28138_c0_g1, DN56432_c0_g7, DN3299_c1_g2, DN102341_c0_g1, and DN56432_c0_g8), and one AHAS (DN12455_c0_g1) unigenes were increased by BR treatment and were decreased by Pcz treatment, respectively, compared to the control.

### 2.4. KEGG-Annotated Genes Involved in Alkaloid Biosynthetic Pathways

In order to identify the synthesis of other alkaloids in *P. ternata*, the keyword of alkaloid was searched in KEGG annotated gene set. It was observed that twenty-five, five, and thirty-eight unigenes were annotated into tropane, piperidine, and pyridine alkaloid biosynthesis (map 00960), indole alkaloid biosynthesis (map 00901), and isoquinoline alkaloid biosynthesis (map 00950) pathway, respectively. In the tropane, piperidine, and pyridine alkaloid biosynthesis, these unigenes were assigned to aspartate aminotransferase, mitochondrial (EC:2.6.1.1, 10 genes), histidinol-phosphate aminotransferase (EC:2.6.1.9, 1 gene), tyrosine aminotransferase (EC:2.6.1.5, 4 genes), hydroxyphenylpyruvate reductase (EC:1.1.1.237, 1 gene), primary-amine oxidase (EC:1.4.3.21, 5 genes), and tropinone reductase I (EC:1.1.1.206, 4 genes) (Figure 4). The four aspartate aminotransferases (DN114811_c0_g1, DN1016_c0_g1, DN15122_c0_g1, and DN69580_c0_g1), two tyrosine aminotransferases (DN114601_c0_g1 and DN50917_c0_g1), one primary-amine oxidase (DN46542_c0_g1), and one tropinone reductase I (DN1777_c0_g1) unigenes were highly expressed in the BR-treated group and were lower expressed in the Pcz-treated group than those in control. Five aromatic-L-amino-acid/L-tryptophan decarboxylase (EC:4.1.1.28 4.1.1.105) unigenes were annotated in the indole alkaloid biosynthesis (Figure 5). The expression level of two aromatic-L-amino-acid/L-tryptophan decarboxylase (DN6751_c0_g1 and DN30743_c1_g1) unigenes were increased by BR treatment and were decreased by Pcz treatment, respectively, compared to the control. In the isoquinoline alkaloid biosynthesis, these unigenes were annotated as aspartate aminotransferase, mitochondrial (EC:2.6.1.1, 10 genes), tyrosine aminotransferase (EC:2.6.1.5, 4 genes), polyphenol oxidase (EC:1.10.3.1, 11 genes), aromatic-L-amino-acid/L-tryptophan decarboxylase (EC:4.1.1.28 4.1.1.105, 5 genes), primary-amine oxidase (EC:1.4.3.21, 5 genes), and norbelladine O-methyltransferase (EC:2.1.1.336, 2 genes) (Figure 6). The three polyphenol oxidases (DN97551_c0_g1, DN17991_c1_g3, and DN22722_c0_g1) and one norbelladine O-methyltransferase (DN1774_c0_g1) unigenes were highly expressed in the BR-treated group and were lower expressed in the Pcz-treated group than those in control.

### 2.5. qRT-PCR Validation of the Sequencing Data

The expression levels of seventeen unigenes were verified by qRT-PCR. The expression trends of nine unigenes were highly consistent with the transcriptome data, and eight unigenes were partially compatible (Appendix A). The R^2^ (COD) and Pearson’s correlation coefficient between the relative expression levels of RNA-Seq and qRT-PCR were 0.6045 and 0.7775, respectively (Appendix A).

## 3. Discussion

Elaborating the enzymatic metabolic biosynthesis of ephedrine in Ephedra species and *Catha edulis* has been very successful. However, there are few studies on the metabolism pathways of ephedrine in *P. ternata*. In addition, the molecular regulatory mechanism of BR as an inducing factor to promote alkaloid accumulation is not clear. In this study, we sequenced the transcriptome of control, BR-treated, and Pcz-treated samples of *P. ternata* bulbil and comparatively analyzed the effects of different treatments on alkaloids metabolic pathways.

BR treatment significantly increased the bulbil yield of *P. ternata*. This is consistent with previous research on *P. ternata* [20], tomato [24], and maize [25]. Maia et al. verified that the application of BR promoted the influx and consequent fixation of CO_2_ and increased the protection of the roots and the uptake of water and nutrients in tomato [26]. Higher carbon assimilation and nutrient supply enhanced the ability of carbon sources to sinks. In addition, BR may be related to cell wall-loosening enzyme activity and autophosphorylation [24,27]. These may be the main reasons that the bulbil yield was increased by BR treatment. Moreover, the bulbil yield was decreased by Pcz treatment, indicating that inhibition of BR biosynthesis negatively affects bulbil growth. A study with similar results showed that BR biosynthesis inhibitor propiconazole caused growth inhibition in soybean [21].

The ephedrine was first isolated from *P. ternata* by Oshio et al. [5]. The low yield of ephedrine has been the main factor limiting the utilization of *P. ternata*. In recent years, transcriptome approaches have become an effective method to explore the regulatory processes of secondary metabolism in plants. In this paper, the expression level of PAL genes was increased by BR treatment and was decreased by Pcz treatment (except for DN2399_c0_g1). Similar results were obtained by Wang et al., who reported that salicylic acid treatment increased the PAL expression level and acteoside content in *Rehmannia glutinosa* [28]. Klempien et al. reported that downregulation of 4CL expression decreased the content of benzyl benzoate and phenylethyl benzoate in petunia flowers [29]. The identification of 4CL genes was highly expressed in BR and Pcz treatments. These results suggested that BR and Pcz promoted the accumulation of benzoate. The expression levels of CHY (DN7851_c0_g1, DN55716_c0_g1, and DN3831_c1_g1) were raised by BR treatment and decreased by Pcz treatment. Similar studies showed that high expression of CHY may be the main reason for the salicylic acid-induced accumulation of benzoic acid and other alkaloids in in vitro cultured *P. ternata* [11]. AHAS may be the enzyme that catalyzes the production of 1-phenylpropane-1,2-dione from benzoic acid. However, these enzymes have not been isolated from plants [10]. KAT catalyzes the generation of benzoyl-CoA from 3-oxo-3-phenylpropionyl-CoA [30]. The expression levels of KAT (DN28138_c0_g1, DN56432_c0_g7, DN3299_c1_g2, DN102341_c0_g1, and DN56432_c0_g8) and AHAS (DN12455_c0_g1) were increased by BR treatment and were decreased by Pcz treatment, suggesting that BR treatment may promote the production of 3-oxo-3-phenylpropionyl-CoA to 1-phenylpropane-1,2-dione in ephedrine biosynthesis. Our results indicated that BR treatment promoted the biosynthesis of ephedrine in the bulbil of *P. ternata* by regulating PAL (DN24895_c0_g1 and DN33061_c1_g1), CHY (DN7851_c0_g1, DN55716_c0_g1, and DN3831_c1_g1), KAT (DN28138_c0_g1, DN56432_c0_g7, DN3299_c1_g2, DN102341_c0_g1, and DN56432_c0_g8), and AHAS (DN12455_c0_g1) genes.

In tropane, piperidine, and pyridine alkaloid biosynthesis, aspartate aminotransferase (EC:2.6.1.1), histidinol-phosphate aminotransferase (EC:2.6.1.9), and tyrosine aminotransferase (EC:2.6.1.5) catalyzes the reversible reaction between L-phenylalanine and phenylpyruvate [31]. Moreover, aspartate aminotransferase (EC:2.6.1.1) has an important role in plant nitrogen and carbon metabolism [32]. Bedewitz et al. reported that silencing of the 4-hydroxyphenylpyruvate aminotransferase gene disrupted the biosynthesis of tropane alkaloids in *Atropa belladonna* by decreasing the amino transfer of phenylalanine to produce phenylpyruvate [33]. The expression levels of aspartate aminotransferase (DN114811_c0_g1, DN1016_c0_g1, DN15122_c0_g1, and DN69580_c0_g1), and tyrosine aminotransferase (DN114601_c0_g1 and DN50917_c0_g1) were enhanced by BR treatment and were reduced by Pcz treatment. Hydroxyphenylpyruvate reductase (EC:1.1.1.237) was demonstrated to catalyze the conversion of phenylpyruvate to phenyllactate, a precursor of tropane alkaloids, and inhibition of hydroxyphenylpyruvate reductase (EC:1.1.1.237) gene expression disrupted tropane alkaloid biosynthesis by reducing phenyllactate levels [34]. We found that BR treatment increased the expression level of hydroxyphenylpyruvate reductase (DN1369_c0_g1). In tropane, piperidine, and pyridine alkaloid biosynthesis, primary-amine oxidase (EC:1.4.3.21) catalyzes N-methyl putrescine to produce 1-methyl prnolinium and catalyzes cadaverine to produce 5-aminopentanal. Our results showed that the expression level of primary-amine oxidase (DN46542_c0_g1) was enhanced by BR treatment and was decreased by Pcz treatment. This result indicated that BR treatment promoted anatalline, anapheline, and pseudo-pelletierine synthesis by improving the primary-amine oxidase. The tropinone reductase I (EC:1.1.1.206) catalyzes tropinone to produce tropine. In root cultures of *Scopolia lurida*, overexpression of ropinone reductase I gene increased the hyoscyamine content by 1.7- to 2.9-fold compared to the control [35]. The expression level of tropinone reductase I (DN1777_c0_g1) was increased by BR treatment and was reduced by Pcz treatment. Our results suggested that BR treatment regulated the expression of aspartate aminotransferase (EC:2.6.1.1), tyrosine aminotransferase (EC:2.6.1.5), primary-amine oxidase (EC:1.4.3.21), and tropinone reductase I (EC:1.1.1.206) genes to promote the accumulation of tropane alkaloids biosynthesis.

Aromatic-L-amino-acid/L-tryptophan decarboxylase (EC:4.1.1.28 4.1.1.105) is a family of enzyme genes that can decarboxylate aromatic amino acid substrates to the corresponding aromatic arylalkylamines [36]. In the indole alkaloid biosynthesis, aromatic-L-amino-acid/L-tryptophan decarboxylase (EC:4.1.1.28 4.1.1.105) catalyze the transfer of L-tryptophan to tryptamine. The expression level of aromatic-L-amino-acid/L-tryptophan decarboxylase (DN6751_c0_g1 and DN30743_c1_g1) was increased by BR treatment and was reduced by Pcz treatment, suggesting that BR treatment may enhance the conversion of L-tryptophan to tryptamine, which promoted the biosynthesis of indole alkaloid.

In addition, polyphenol oxidase (EC:1.10.3.1) and aromatic-L-amino-acid/L-tryptophan decarboxylase (EC:4.1.1.28 4.1.1.105) catalyze the production of dopamine from L-tyrosine [36]. Our results showed that the expression level of polyphenol oxidase (DN97551_c0_g1, DN17991_c1_g3, and DN22722_c0_g1) was improved by BR treatment and was reduced by Pcz treatment. Norbelladine O-methyltransferase (EC:2.1.1.336) catalyzes the formation of 4′-methylnorbelladine from norbelladine, which has been suggested to derive all amaryllidaceae alkaloids. The expression level of norbelladine O-methyltransferase (DN1774_c0_g1) was increased by BR treatment and was decreased by Pcz treatment. Li et al. reported that the transcript level of the norbelladine O-methyltransferase gene was positively correlated with the amaryllidaceae alkaloid content in *Lycoris radiata* [37]. These results indicated that BR treatment might regulate the expression of aspartate aminotransferase (EC:2.6.1.9), tyrosine aminotransferase ((EC:2.6.1.5), polyphenol oxidase (EC:1.10.3.1), aromatic-L-amino-acid/L-tryptophan decarboxylase (EC:4.1.1.28 4.1.1.105), primary-amine oxidase (EC:1.4.3.21), and norbelladine O-methyltransferase (EC:2.1.1.336) genes to promote the accumulation of isoquinoline alkaloid biosynthesis.

After BR or Pcz treatments, the transcriptome data were used to analyze the expression patterns of unigenes annotated as alkaloid biosynthesis. Several genes involved in the ephedrine alkaloid, tropane, piperidine, and pyridine alkaloid, indole alkaloid, and isoquinoline alkaloid biosynthesis were up-regulated in BR treatment and were down-regulated in Pcz treatment. Moreover, the total alkaloid content and bulbil yield of *P. ternata* was increased by 90.87% and 29.67% under BR treatment compared to the control. The results were consistent with the previous hypothesis that BR treatment increased the alkaloid content by enhancing the alkaloid biosynthesis. We identified several highly expressed genes in alkaloid biosynthesis of *P. ternata* treated with BR treatment, and these data contribute to the further study of the molecular mechanism of BR-promoted alkaloid biosynthesis.

## 4. Materials and Methods

### 4.1. Plant Material

Seeds bulbs of *P. ternata* were sown in pots containing humus soil. After three-leaf expanded, the control, BR-treated, and Pcz-treated plants were supplemented through the foliage with distilled water, 0.1 mg L^−1^ BR (Anpel Laboratory Technologies (Shanghai) Inc., Shanghai, China), and 1 μM Pcz (Aladdin Reagent (Shanghai) Co., Ltd., Shanghai, China) every day, respectively. Tween 20 (0.02%, *v/v*, Beijing Kulaibo Technology Co., Ltd., Beijing, China), as a surfactant, was mixed with the distilled water, BR, and Pcz solution. Pots were maintained in a completely randomized design placed under indoor conditions with four replicates for each treatment. Tuber and bulbil samples were harvested at the bulbil expansion stage and snap-frozen using liquid nitrogen until RNA extraction and determination of physiological indexes.

### 4.2. Bulbil Yield and Total Alkaloid Content

Each treatment had four biological replicates with ten random bulbils per replicate. The bulbil yield was expressed as g plant^−1^. Total alkaloid was determined using the method of Guo et al. [20]. Dried samples were extracted with 0.5 mL 25% ammonium hydroxide and 5.0 mL chloroform at 37 °C for 1 h. The reaction mixture contained 200 μL extract solution, 5 mL citric acidsodium citrate (pH 5.4), 0.5 mL bromothymol blue (0.1%) and 4.8 mL chloroform. After one hour, the chloroform layer was measured in a spectrophotometer at 416 nm and total alkaloid content was expressed as mg g^−1^ dried weight (DW).

### 4.3. Total RNA Isolation, cDNA Library Construction, and Sequencing

For RNA-Seq and qRT-PCR, RNA samples were prepared using three replicates of bulbil from control, BR-treated, and Pcz-treated groups. We used the RNA extraction kit (Cowin Biosciences, Beijing) to extract total RNA and checked quality and quantity of RNA by NanoDrop2000, Agilent2100 Nano, and RNase-free agarose gel electrophoresis. The Illumina Truseq^TM^ RNA sample preparation kit technology was used to prepare RNA-Seq libraries, and the libraries were sequenced on an Illumina Nova seq 6000 platform.

### 4.4. Transcriptome Assembly and Functional Annotation

After sequencing, the clean data were obtained by using the software SeqPrep (https://github.com/jstjohn/SeqPrep, accessed on 22 December 2020) and sickle (https://github.com/najoshi/sickle, accessed on 22 December 2020) to remove low-quality reads, adapters, any anonymous nucleotides greater than 10%, and linker sequences from the raw data. Clean data assembly was performed using Trinity (Version v2.8.5) software as described in Grabherr et al., and the assembly results were evaluated and optimized using transrate (Version v1.0.3) and busco (Version 3.0.2) software [38]. We used software of HMMER (Version 3.1b2) and Diamond (Version v0.8.37.99) to compare the assembled unigenes database with the Pfam (Version 33.1), swiss-prot (Version 2020.06), clusters of orthologous groups of proteins (COG, Version 2020.06), and national center for biotechnology information non-redundant protein (NR, Version 2020.06) and obtained functional annotations of unigenes by comparing the protein with highest sequence similarity. In addition, the functional annotations and biological pathway analysis of unigenes were also performed in Kyoto encyclopedia of genes and genomes (KEGG, Version 2020.07) database and gene ontology (GO, Version 2020.0628) database using Kobas (Version 2.1.1) and Blast2gO (Version 2.5.0) software.

### 4.5. Differentially Expressed Analysis

The DEGs, after BR and Pcz treatments, were analyzed using DEseq2 (Version 1.38.0) as described by Love et al. [39]. The expression levels were quantified based on transcripts per million reads (TPM) values of the gene. We used a *p* < 0.05 and |log2 (fold change) | ≥ 2 as the threshold for determining the significant differences between treatment and control samples. The significant enrichment terms of DEGs on the GO and KEGG database were calculated with adjusted *p*-values (FDR) < 0.05 using the Majorbio I-Sanger cloud platform (www.majorbio.com, accessed on 22 December 2020) and Goatools (Version 0.6.5) software. Subsequently, we screened unigenes participating in alkaloid biosynthesis from the functional annotation results and generated heatmaps of expression levels of these genes in control, BR-treated, and Pcz-treated groups.

### 4.6. qRT-PCR Analysis

Seventeen differentially expressed unigenes were selected for qRT-PCR analysis. The reverse transcription kit (Hiscript III RT supermix for qPCR (+gDNA wiper)) (Vazyme, Nanjing, China) reverse transcribed 1 μg of total RNA into cDNA for qPCR. The qRT-PCR reactions were carried out using a 20 μL reaction volume containing Chamq universal SYBR qPCR master mix (Vazyme, Nanjing, China), primer (Sangon Biotech (Shanghai) Co., Ltd., Shanghai, China), cDNA template, and ddH_2_O (Vazyme, Nanjing, China) and were performed on a LightCycle 96 system (Roche, Switzerland). All primers of unigenes were designed online by the Integrated DNA Technologies website (Appendix A). The 18s gens of *P. ternata* was used as a reference gene. The relative expression levels of unigenes were calculated by the 2^−ΔΔCT^ method using the 18S rRNA gene of *P. ternata* as a reference gene [3].

### 4.7. Statistical Analysis

The physiological data had four biological replicates, expressed as mean ± standard error, and were statistically analyzed using the software Statistical Package for Social Science (SPSS, version 26.0). Homogeneity of variance was tested using the Levene test before analysis. The analysis of variance (ANOVA) followed by Tukey’s test was used to determine the significant difference (*p* < 0.05).

## Figures and Tables

**Figure 1 ijms-23-10898-f001:**
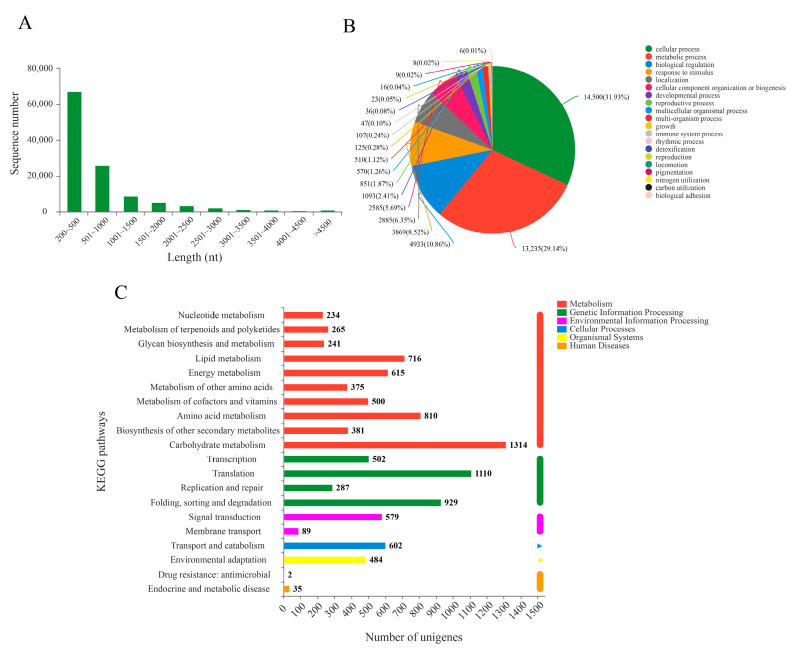
Functional annotations of the unigenes of the *P. ternata* bulbil transcriptome. (**A**) length distribution of assembled unigenes; (**B**) GO function annotation of *P. ternata*; (**C**) KEGG function annotation of *P. ternata*.

**Figure 2 ijms-23-10898-f002:**
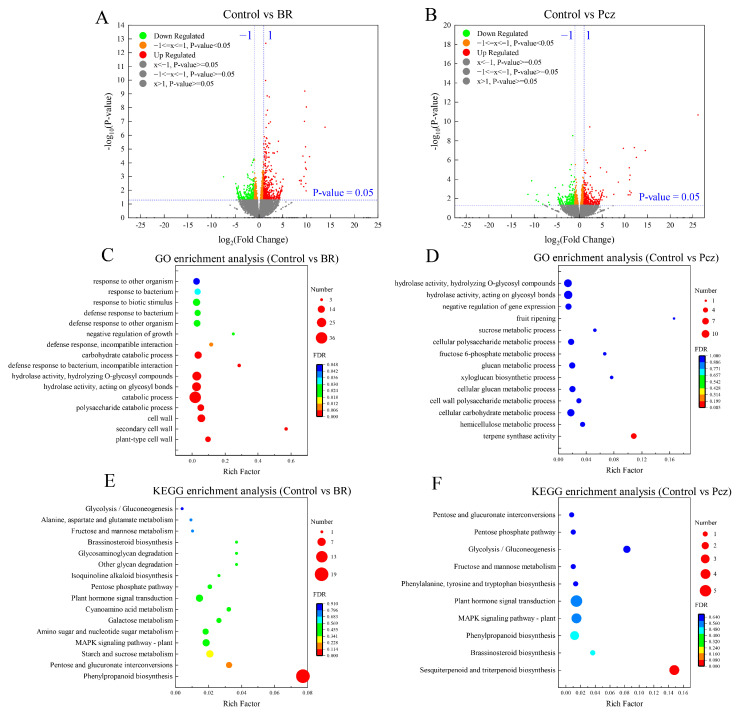
Identification and functional information of differential expression genes (DEGs) among control, BR, and Pcz treatments in *P. ternata*. (**A**) volcano of DEGs in control and BR groups; (**B**) volcano of DEGs in control and Pcz groups; (**C**) GO enrichment analyses with the DEGs generated in control and BR groups; (**D**) GO enrichment analyses with the DEGs generated in control and Pcz groups; (**E**) KEGG enrichment analyses with the DEGs generated in control and BR groups; (**F**) KEGG enrichment analyses with the DEGs generated in control and Pcz groups.

**Figure 3 ijms-23-10898-f003:**
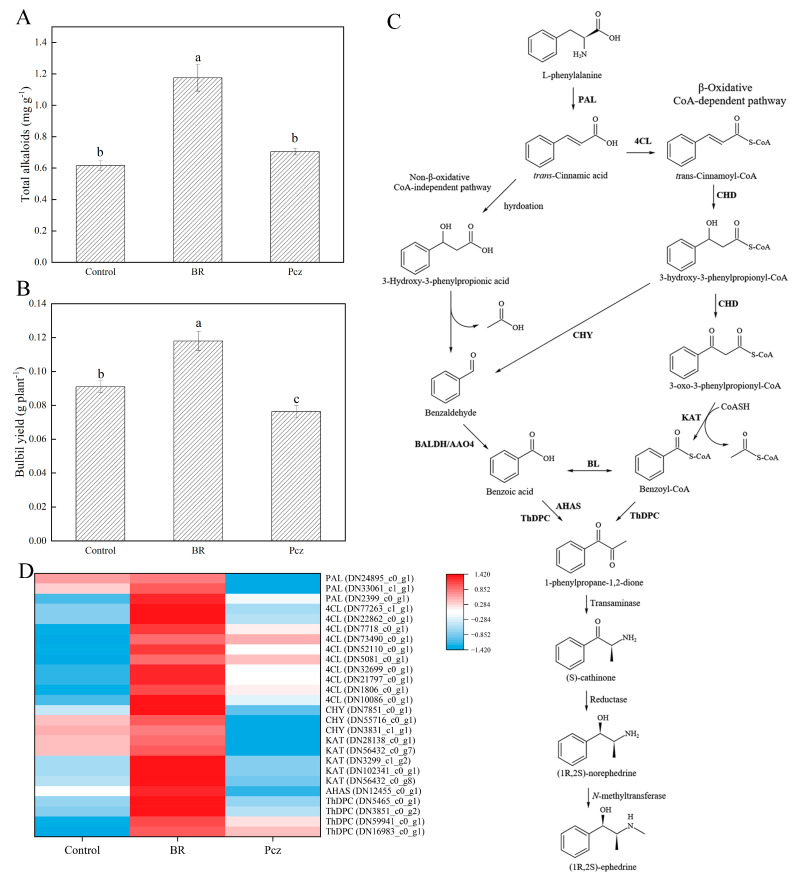
Effects of BR and Pcz treatments on the total alkaloid contents (**A**) and bulbil yield (**B**). The bars with different letters are significantly different from each treatment (*p* < 0.05). Values are means of four replicates ± SE. Proposed biosynthetic routes of ephedrine in *P. ternata* (**C**). Differentially expressed levels of unigenes related to ephedrine biosynthesis (**D**). Changes in expression level are represented by a change in color; blue indicates a lower expression level, whereas red indicates a higher expression level. All data shown reflect the average mean of three biological replicates.

**Figure 4 ijms-23-10898-f004:**
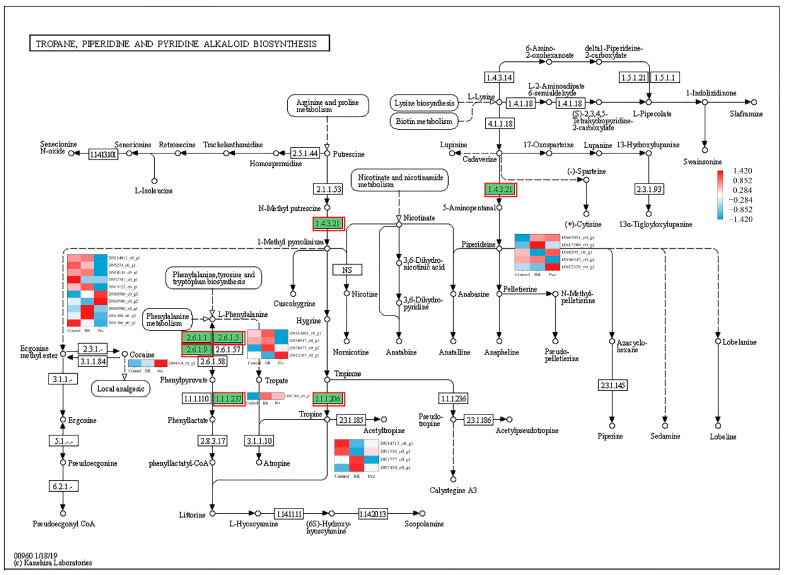
Differential expression levels of unigenes related to tropane, piperidine, and pyridine alkaloid biosynthesis identified by KEGG annotation. Changes in expression level are represented by a change in color; blue indicates a lower expression level, whereas red indicates a higher expression level. All data shown reflect the average mean of three biological replicates.

**Figure 5 ijms-23-10898-f005:**
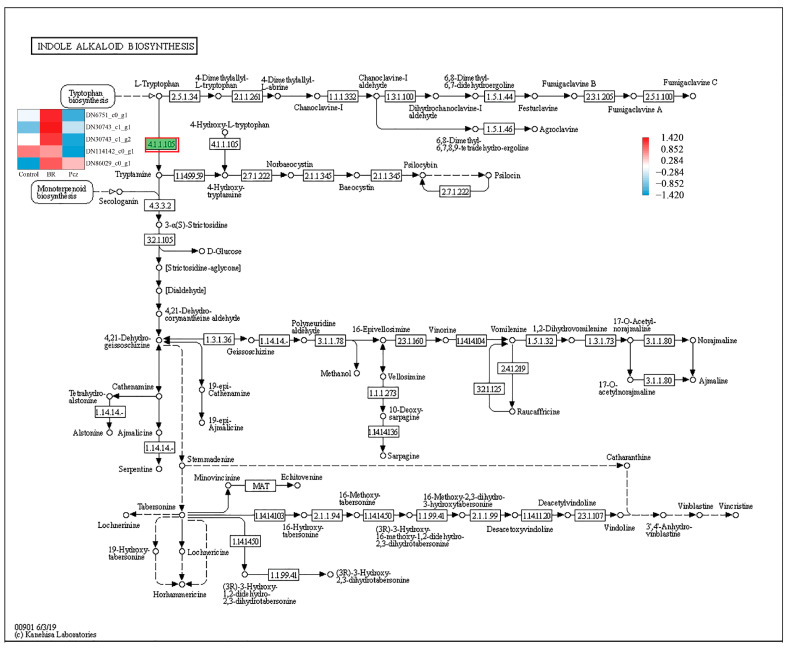
Differential expression levels of unigenes related to indole alkaloid biosynthesis identified by KEGG annotation. Changes in expression level are represented by a change in color; blue indicates a lower expression level, whereas red indicates a higher expression level. All data shown reflect the average mean of three biological replicates.

**Figure 6 ijms-23-10898-f006:**
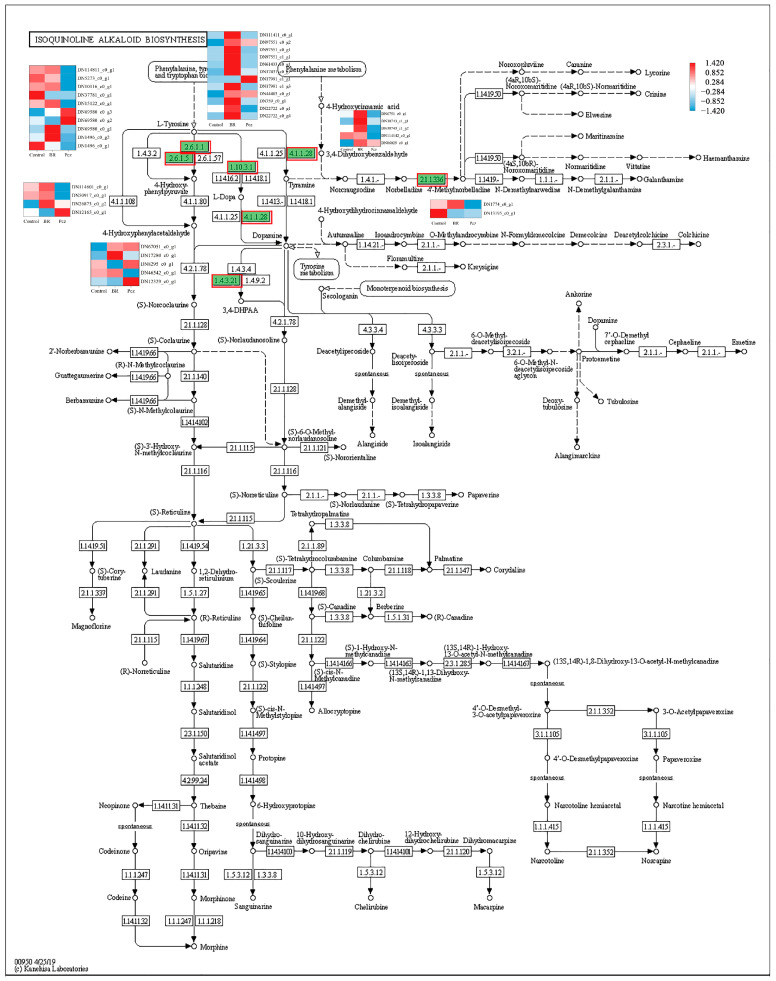
Differential expression levels of unigenes related to isoquinoline alkaloid biosynthesis identified by KEGG annotation. Changes in expression level are represented by a change in color; blue indicates a lower expression level, whereas red indicates a higher expression level. All data shown reflect the average mean of three biological replicates.

## Data Availability

Raw reads were deposited in the NCBI SRA database.

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
