# Peer review of "Transcriptome Analysis Reveals an Essential Role of Exogenous Brassinolide on the Alkaloid Biosynthesis Pathway in Pinellia Ternata"

_ijms, 2022, doi:10.3390/ijms231810898_

Round 1
Reviewer 1 Report
Dear Authors,
The manuscript “Transcriptome analysis reveals an essential role of exogenous brassinolide on the alkaloid biosynthesis pathway in Pinellia ternata” analyses the effects of Brassinolide (BR) and BR biosynthesis inhibitor (propiconazole, Pcz) treatments on the alkaloid biosynthesis in bulbil of P. ternata. The results showed that BR treatment increased the alkaloid content by enhancing the alkaloids biosynthesis.
In my opinion this work is very interesting, well structured and the methods used are correct.
There are no comments.
Just the following minor changes are suggested:
Page 1, line 32: Author(s) of species name must be provided when the scientific name of any plant species is first mentioned (i.e. Pinellia ternata/ Pinellia ternata (Thunb.) Druce. Correct throughout the manuscript.
Author Response
Response to Reviewer 1 Comments
Point 1: Page 1, line 32: Author(s) of species name must be provided when the scientific name of any plant species is first mentioned (i.e. Pinellia ternata/ Pinellia ternata (Thunb.) Druce. Correct throughout the manuscript.
Response 1: Yes, we revised the species name in the revised paper. Thanks! See page 1, lines 13 and 33, please.

Reviewer 2 Report
The manuscript "Transcriptome analysis reveals an essential role of exogenous brassinolide on the alkaloid biosynthesis pathway in Pinellia ternata" by C. Guo et al. is devoted to the study of molecular mechanisms mediated by brassinolide on the biosynthesis of secondary metabolites in the medical plant Pinellia ternata.
The transcriptomes of plants treated by brassinolide and a brassinolide biosynthesis inhibitor, as well as the total alkaloid content, were analyzed by the authors, and compared to untreated (control) plants. The presented data indicates that brassinolide treatment increases alkaloid content by enhancing alkaloid biosynthesis. The work is well-written, easy to read and illustrated. I believe it could be accepted for publication after revision. Some critical points are listed below.
1. It would be better if the structure of the discussed compounds were depicted. I mean brassinolide and propiconazole, first of all.
2. Only total alkaloid content of the plant bulbils was evaluated by the authors. It would be more informative and interesting to compare the content of different individual compounds of the class. It is possible by HPLC-MS analysis of the extracts or using other analytical methods.
3. Minor revisions needed:
lines 56, 58, 61: Ephedra species - "species" should not be itallic;
line 67: trans-Cinnamic acid - "Cinnamic acid" should not be itallic;
line 179: 18s gene - 18S rRNA gene.
Author Response
Response to Reviewer 2 Comments
Point 1: It would be better if the structure of the discussed compounds were depicted. I mean brassinolide and propiconazole, first of all.
Response 1: Yes, we added the structural descriptions of brassinolide and propiconazole in revised paper. Thanks! See page 2, lines 83 and 97, please.
Point 2: Only total alkaloid content of the plant bulbils was evaluated by the authors. It would be more informative and interesting to compare the content of different individual compounds of the class. It is possible by HPLC-MS analysis of the extracts or using other analytical methods.
Response 2: Sorry, we did not measure individual compounds of the alkaloid in bulbil. The proposal is excellent and gives us a good idea. Next, we will further investigate the effect of brassinolide and brassinolide biosynthesis inhibitor treatments on the content of individual alkaloid compounds by metabolomics (HPLC-MS analysis). Thanks again.
Point 3: Minor revisions needed:
3.1: lines 56, 58, 61: Ephedra species - "species" should not be itallic;
Response 3.1: Yes, we revised it in revised paper. See page 2, lines 57, 59, and 62, please.
3.2: line 67: trans-Cinnamic acid - "Cinnamic acid" should not be itallic;
Response 3.2: Yes, we revised it in revised paper. See page 2, line 68, please.
3.3: line 179: 18s gene - 18S rRNA gene.
Response 3.3: Yes, we revised it in revised paper. See page 4, line 182, please.

Round 2
Reviewer 2 Report
Structures, not “structural descriptions” of the discussed compounds should be added to the manuscript.
Author Response
Response to Reviewer 2 Comments
Point 1: Structures, not “structural descriptions” of the discussed compounds should be added to the manuscript.
Response 1: Yes, we added the structures of brassinolide and propiconazole in the revised paper. Please let me know if it still has any questions. Thanks! “Brassinolide ((22R,23R,24S)-2α,3α,22,23-tetrahydroxy-24-methyl-B-homo-7-oxa-5α-cholestan-6-one, BR) is a plant hormone that plays a key role in plant growth and development.” “In addition, propiconazole (1-[ [2-(2,4dichlorophenyl)-4-propyl-1,3-dioxolan-2-yl]methyl]-1,2,4-triazole, Pcz) is an economical and specific inhibitor of BR biosynthesis in Arabidopsis, maize, and soy-bean”
